# FIGO 2018 Staging for Cervical Cancer: Influence on Stage Distribution and Outcomes in the 3D-Image-Guided Brachytherapy Era

**DOI:** 10.3390/cancers12071770

**Published:** 2020-07-02

**Authors:** Kento Tomizawa, Takuya Kaminuma, Kazutoshi Murata, Shin-ei Noda, Daisuke Irie, Takuya Kumazawa, Takahiro Oike, Tatsuya Ohno

**Affiliations:** 1Department of Radiation Oncology, Gunma University Graduate School of Medicine, 3-39-22, Showa-machi, Maebashi, Gunma 371-8511, Japan; m2020015@gunma-u.ac.jp (K.T.); cami_taku@gunma-u.ac.jp (T.K.); kazutoshi.m@gunma-u.ac.jp (K.M.); daisuke_i@gunma-u.ac.jp (D.I.); m1820021@gunma-u.ac.jp (T.K.); tohno@gunma-u.ac.jp (T.O.); 2Gunma University Heavy Ion Medical Center, 3-39-22, Showa-machi, Maebashi, Gunma 371-8511, Japan; 3Department of Radiation Oncology, Saitama Medical University International Medical Center, 1397-1, Yamane, Hidaka, Saitama 350-1298, Japan; nodashin@saitama-med.ac.jp

**Keywords:** uterine cervical cancer, radiotherapy, image-guided brachytherapy, outcomes, FIGO 2018 staging system

## Abstract

Recent widespread use of three-dimensional image-guided brachytherapy (3D-IGBT) has improved radiotherapy outcomes of cervical cancer dramatically. In 2018, the International Federation of Gynecology and Obstetrics (FIGO) staging system for cervical cancer was revised. However, the influence of the revisions on the stage distribution and outcomes of cervical cancers treated with 3D-IGBT remains unclear. Here, we retrospectively analyzed 221 patients with cervical squamous cell carcinoma treated with definitive radiotherapy using 3D-IGBT (median follow-up, 60 months). The stage distribution and outcomes were compared between the 2009 and 2018 schemas. Stage migration occurred in 52.9% of the patients. Patients classified with the 2018 criteria as stage IIIC_r_ had the highest proportion (43.8%) of migration, and were mainly from the 2009 stages IIB and IIIB. The 2009 and 2018 schemas showed comparable performance at stratifying 5-year overall survival (OS) and 5-year progression-free survival (PFS) for patients in stages IB–IVA. The 2018 criteria effectively stratified 5-year OS and PFS in the stage III substages. The 5-year OS and PFS for stage IIIC1_r_ patients varied according to tumor T stage. These data provide evidence for the utility of the revised 2018 FIGO staging system in the clinical management of cervical cancers in the 3D-IGBT era.

## 1. Introduction

Cervical cancer causes more than 0.3 million deaths worldwide annually and its mortality ranks fourth among all cancers [1]. Radiotherapy plays a pivotal role in definitive treatment for locally advanced cervical cancers [2]. Definitive radiotherapy consists of external beam radiotherapy (EBRT) and brachytherapy [3]. For the latter, three-dimensional image-guided brachytherapy (3D-IGBT) using magnetic resonance imaging (MRI) or computed tomography (CT) has become widespread in recent years [4,5]. The introduction of 3D-IGBT technologies to the practice of definitive radiotherapy has improved the outcomes of cervical cancers dramatically [6,7].

Historically, the International Federation of Gynecology and Obstetrics (FIGO) staging guidelines for cervical cancer had allowed assessment of the extent of disease to be mainly based on clinical examination including bimanual pelvic examination, cystoscopy, and colonoscopy [8,9]. This was to enable the comparison of outcomes across the resource-limited countries where this disease occurs most commonly [10]. In 2018, FIGO released revised staging guidelines that for the first time allowed the use of imaging modalities and pathologic assessment for staging [11]. A major change in the FIGO 2018 staging system is that stage IB patients are classified into three substages by tumor size: stage IB1 (<2 cm), stage IB2 (2–3.9 cm), and stage IB3 (≥4 cm). Another major change is the incorporation of nodal status into stage III disease, where patients positive for metastasis to pelvic lymph nodes and to paraaortic lymph nodes (PALNs) are classified into stages IIIC1 and IIIC2, respectively, with the notation r (imaging) or p (pathology) to indicate the modality used for diagnosis. To date, two studies have reported the prognostic performance of the FIGO 2018 staging system using populations extracted from the national databases of the United States [10,12]. In these reports, however, the study populations are highly heterogeneous in terms of clinical management and data collection period and include a considerable number of patients registered in the pre-3D-IGBT era. Therefore, the influence of the revisions to the FIGO 2018 staging system on the stage distribution and outcomes of cervical cancers treated with definitive radiotherapy using 3D-IGBT remains unclear. To address this issue, we assembled a study cohort that comprised solely of cases treated with definitive radiotherapy using 3D-IGBT and compared the stage distribution and outcomes between the FIGO 2009 and FIGO 2018 staging systems.

## 2. Results

This study included 221 patients with squamous cell carcinoma of the uterine cervix (Sq-UCC). The study design is described in Section 4.1. Median follow up was 60.0 months (range, 4.0–134.7 months). Median age was 61 years (range, 27–91). Sixty-two percent of the patients received chemotherapy concomitantly with radiotherapy; 78.2% of the patients received ≥4 courses, whereas the smaller number of courses were mainly due to hematologic toxicities (Table 1).

Firstly, we compared stage distribution between the FIGO 2009 and FIGO 2018 staging systems (Figure 1). Overall, stage migration was observed in 117/221 (52.9%) of the patients. With the FIGO 2009 criteria, the study cohort predominantly comprised stage IIB and IIIB patients (37.5% and 33.0%, respectively; Figure 1A). By contrast, with the FIGO 2018 criteria, stage IIIC_r_ patients represented the highest proportion (43.8%), with a decrease in the proportions of IIB and IIIB patients (Figure 1A); these stage IIIC_r_ patients mainly migrated from former stage IIB (56.7%) or stage IIIB (30.9%; Figure 1B,C). Besides, the distribution of stage IB patients showed a shift to the right (from FIGO 2009 to FIGO 2018) based on the changes in the tumor size criteria, as expected (Figure 1A). These data indicate that the revisions to the FIGO staging criteria, particularly the introduction of stage IIIC to the FIGO 2018 schema, markedly changed the stage distribution of Sq-UCC patients treated with definitive radiotherapy.

Secondly, we compared the prognostic performance of the FIGO 2009 and FIGO 2018 staging systems on patients with stage IB, II, III, and IVA disease. In both the FIGO 2009 and FIGO 2018 staging systems, there was a significant trend toward worse overall survival (OS) for advanced-stage patients (*p* = 0.032 and 0.0021, respectively; Figure 2A,B). The 5-year OS according to the FIGO 2009 schema was 90.5% (95% confidence interval: CI, 73.3–96.8%) for stage IB patients, 86.8% (95% CI, 77.4–92.5%) for stage II patients, 67.0% (95% CI, 54.8–76.6%) for stage III patients, and 73.0% (95% CI, 42.9–88.9%) for stage IVA patients (Table 2). Meanwhile, the 5-year OS according to the FIGO 2018 schema was 86.7% (95% CI, 64.1–95.5%) for stage IB patients, 88.9% (95% CI, 76.9–94.9%) for stage II patients, 73.7% (95% CI, 64.3–81.0%) for stage III patients, and 73.0% (95% CI, 42.9–88.9%) for stage IVA patients (Table 2). The 5-year OS for each stage did not differ significantly between the FIGO 2009 and FIGO 2018 schemas (*p* = 0.65, 0.60, 0.34, and 1.0 for stages IB, II, III and IVA, respectively; Table 2).

In both the FIGO 2009 and FIGO 2018 staging systems, there was a significant trend toward worse progression-free survival (PFS) for advanced-stage patients (*p* = 0.0002 and 0.0010, respectively; Figure 3A,B). The 5-year PFS according to the FIGO 2009 schema was 89.8% (95% CI, 71.7–96.6%) for stage IB patients, 79.0% (95% CI, 69.1–86.1%) for stage II patients, 59.3% (95% CI, 47.3–69.4%) for stage III patients, and 58.3% (95% CI, 29.1–78.9%) for stage IVA patients (Table 3). Meanwhile, the 5-year PFS according to the FIGO 2018 schema was 85.4% (95% CI, 61.1–95.0%) for stage IB patients, 85.2% (95% CI, 73.5–92.0%) for stage II patients, 64.3% (95% CI, 54.8–72.3%) for stage III patients, and 58.3% (95% CI, 29.1–78.9%) for stage IVA patients (Table 3). The 5-year PFS for each stage did not differ significantly between the two staging systems (*p* = 0.63, 0.34, 0.48, and 1.0 for stages IB, II, III, and IVA, respectively; Table 3). Together, these data indicate that the prognostic performance of the FIGO 2009 and 2018 criteria are broadly consistent in Sq-UCC patients treated with definitive radiotherapy.

Thirdly, we analyzed the prognostic performance of the FIGO 2018 staging system on stage III disease, the stage that underwent the most significant revisions. Here, the survival of stage IIIA and IIIB patients was analyzed in combination as there were only two stage IIIA patients. There was a significant trend toward worse OS for advanced-stage patients (*p* = 0.039; Figure 4A). The 5-year OS was 85.2% (95% CI, 60.8–94.9%) for stage IIIA–B patients, 75.9% (95% CI, 63.1–84.8%) for stage IIIC1_r_ patients, and 60.8% (95% CI, 40.3–76.1%) for stage IIIC2_r_ patients (Table 4). There was also a significant trend toward worse PFS for advanced-stage patients (*p* = 0.028; Figure 4B). The 5-year PFS was 80.9% (95% CI, 56.8–92.3%) for stage IIIA–B patients, 64.6% (95% CI, 51.6–74.9%) for stage IIIC1_r_ patients, and 51.7% (95% CI, 32.5–67.9%) for stage IIIC2_r_ patients (Table 4). These data indicate that the FIGO 2018 staging system effectively stratifies the outcomes of stage III Sq-UCC patients treated with definitive radiotherapy.

Lastly, we analyzed the influence of the T stage (based on the classification by the Union for International Cancer Control, 7th edition) on the outcomes of stage IIIC1_r_ patients because the stage IIIC1_r_ population, defined as being positive for pelvic node involvement, can exhibit tumor size heterogeneity. There was a significant trend toward worse OS for patients with advanced T stage (*p* = 0.013; Figure 5A). The 5-year OS was 100.0% for T1b tumors, 82.5% (95% CI, 59.7–93.0%) for T2 tumors, and 65.2% (95% CI, 45.4–79.2%) for T3 tumors (Table 5). There was also a significant trend toward worse PFS for patients with advanced T stage (*p* = 0.038; Figure 5B). The 5-year PFS was 100.0% for T1b tumors, 65.6% (95% CI, 42.8–81.1%) for T2 tumors, and 54.7% (95% CI, 36.0–70.0%) for T3 tumors (Table 5). These data indicate that the outcomes of stage IIIC1_r_ Sq-UCC patients treated with definitive radiotherapy differ by tumor T stage. A trend toward worse OS and PFS for advanced T stage cases was also observed for stage IIIC2 patients, although it did not reach statistical significance, probably due to the small number of patients (*n* = 30; Appendix A).

## 3. Discussion

This is the first report on the influence of the 2018 revisions to the FIGO staging system on the stage distribution and outcomes of cervical cancer patients treated with definitive radiotherapy in the 3D-IGBT era. The 5-year OS and PFS were effectively stratified by the revised staging criteria, indicating its utility in the practice of definitive radiotherapy for cervical cancers.

To date, three large-scale studies have compared the prognostic performance of the 2009 and 2018 versions of the FIGO staging system using cervical cancer cases registered to The National Cancer Database (*n* = 62,212) [10], The Surveillance, Epidemiology, and End Results Program (*n* = 20,642, stages IB and III only) [12], and Washington University (*n* = 1282) [13]. Although the study populations were large, these studies had shortcomings. Firstly, the diagnostic procedure for lymph node involvement was not standardized in terms of pathology, CT, MRI, and/or ^18^F-fluorodeoxyglucose positron emission tomography (FDG-PET), making it impossible to discern stages IIIC_r_ and IIIC_p_. Secondly, the study populations were heterogeneous in terms of treatment modality (i.e., surgery, radiotherapy, and/or chemotherapy). Thirdly, the three study populations all contained old cases, some from 1988 [12], 1997 [13], and 2004 [10]. In contrast to these studies, our study has the strength that clinical management (i.e., diagnosis, treatment, and follow-up) was standardized throughout the study period, with all the participants being treated using 3D-IGBT.

A major difference in the results between the previous studies and the present study is the prognostic performance of the FIGO 2018 schema for stage III patients. In the previous studies, stratification of survival of stage III patients did not correlate with the order of advancing substage due to the better survival of stage IIIC1 patients than stage IIIA–B patients [10,12,13]. In the present study, by contrast, the survival of stage III patients was effectively stratified in the order of substage. The difference can be attributed to the difference in survival of stage IIIA–B patients, which was approximately 30% greater in our study (approximately 40–50% versus 85% for 5-year OS in the previous studies and the present study, respectively [10,12], and approximately 50% versus 85% for 5-year PFS in the previous studies and the present study, respectively [13]). On the other hand, the survival of stage IIIC patients was comparable between the previous studies and the present study. These data underline the impact of advances in treatment protocols for stage IIIA–B cervical cancers (namely, concurrent chemoradiotherapy) in recent decades, to which 3D-IGBT contributed to through improved control of locally advanced tumors. Another possible reason for the differences may be the unstandardized diagnostic procedures in the previous studies [10,12]; for example, the routine practice of FDG-PET screening in those studies could have upstaged stage IIIA–B cases to stage IIIC due to the detection of occult lymph node involvement. The exclusion of adenocarcinoma cases from the cohort of the present study should have improved the survival rate because the prognosis for adenocarcinoma is significantly worse than that of squamous cell carcinoma [14]. Nevertheless, this is unlikely to solely account for the 30% increase in 5-year survival of stage IIIA–B patients seen in the present study, because the difference in 5-year survival for stage III cervical cancers between adenocarcinoma and squamous cell carcinoma is only around 13–18% [15,16] and adenocarcinoma comprises approximately 10–20% of all cervical cancers [14].

Previous studies showed that the 5-year OS and PFS for stage IIIC1 patients are stratified by tumor T stage [10,12]. These findings were validated in our cohort with pelvic lymph node involvement determined solely by imaging (stage IIIC1_r_). These data underscore the need to further optimize treatment strategy based on the patterns of failure after initial treatment.

As shown in Figure 2 and Figure 3, Kaplan–Meier survival estimates for OS and PFS for stage IB patients were worse in 2018 classification compared with 2009 classification despite upward stage migration in 9 patients. In fact, in our dataset, the 9 patients experiencing upward stage migration had no event for death throughout the follow-up, leading to worse OS and PFS estimates for stage IB patients in 2018 classification. Nevertheless, we think that we cannot draw a solid conclusion on the cause of this phenomenon because the number of patients in our dataset was small (*n* = 33 and 24 in 2009 and 2018 classifications, respectively). A further study employing a larger population is needed.

The present study has the following limitations. Firstly, this was a single-center retrospective study. Secondly, we were unable to analyze the FIGO 2018 stage IB substages due to the small number of cases. Thirdly, adenocarcinoma cases were not analyzed. Fourthly, we were not able to analyze the influence of the number of lymph node metastases on prognosis in 2018 classification due to the small-sized cohort, warranting further research using a larger population. Lastly, we were not able to perform FDG-PET for routine post-radiotherapy surveillance due to socioeconomic reasons. Alternatively, FDG-PET/CT was performed for the cases with clinical or radiological suspicion of recurrence elucidated by routine bimanual examination, CT, and/or MRI.

## 4. Materials and Methods

### 4.1. Study Design

This retrospective study enrolled patients who met the following inclusion criteria: (i) newly diagnosed and pathologically confirmed Sq-UCC; (ii) staged as IB–IVA based on the FIGO 2009 staging criteria; (iii) treated at the Gunma University Hospital (Maebashi, Gunma, Japan) from 2009–2017; and (iv) treated with definitive radiotherapy using 3D-IGBT. Adenocarcinoma was excluded from this study to increase the uniformity of the cohort because adenocarcinoma behaves differently from squamous cell carcinoma in terms of disease progression pattern and response to treatment [14,17], and the proportion of adenocarcinoma among all cervical cancer cases differs by FIGO stage [12].

Data on patient age, FIGO stage (based on the 2009 criteria), tumor, node and metastasis (TNM) factors (based on the classification by the Union for International Cancer Control, 7th edition), and treatment were collected from medical records. The patients were restaged based on the FIGO 2018 staging criteria and then, stage distribution and outcomes were compared between the FIGO 2009 and FIGO 2018 schemas. OS [10] and PFS [13] were used as outcome endpoints.

This study was approved by the institutional ethical review committee of the Gunma University Hospital (approval number: H2019-226). The requirement for written consent from participants was waived due to the retrospective observational nature of the study.

### 4.2. Diagnostic Modality for Staging

For staging, the patients received the following examinations: (a) bimanual pelvic examination, (b) cystoscopy, (c) colonoscopy, (d) CT from the neck to the pelvis, (e) MRI of the pelvis, and (f) FDG-PET from the neck to the pelvis. All the pre-treatment examinations (i.e., a–f) were performed at Gunma University Hospital. The FIGO 2009 staging used the findings from a–c. The FIGO 2018 staging used the findings from a–f. This led to the notation for stage IIIC as stage IIIC_r_ in all patients.

### 4.3. Treatment

Definitive (chemo) radiotherapy was performed as described previously [7]. Briefly, radiotherapy consisted of EBRT and CT-based 3D-IGBT. EBRT was performed at 2 Gy per fraction, five fractions per week. Whole pelvic irradiation was delivered to a total of 50 Gy; a central shielding was inserted at 20 Gy for stage IB1–II tumors of ≤4 cm, at 40 Gy for bulky tumors, or at 30 Gy for the other cases, followed by a 6–10 Gy boost to metastatic pelvic lymph nodes. For patients with PALN metastasis, prophylactic irradiation to the PALN regions with 40 Gy, followed by a 10–16 Gy boost to metastatic PALNs, was added to regular whole pelvic irradiation.

CT-based 3D-IGBT was performed using a high-dose rate ^192^Ir remote afterloading system (microSelectron, Elekta, Stockholm, Sweden) with a tandem and ovoid applicator or a vaginal cylinder [7]. A total of 24 Gy was delivered in four fractions, one fraction per week. Trocar point needles (Nucletron, Elekta, Stockholm, Sweden) were used in combination with the applicators for bulky and/or irregularly shaped tumors.

Cisplatin-based chemotherapy was administered concomitantly with radiotherapy for patients with stage IB2–II tumors of >4 cm, stage III–IVA tumors of any size, or lymph node metastasis [7]. Chemotherapy was not performed for patients aged >75 years, or those who had severe comorbidities (e.g., renal dysfunction, severe diabetes, or ischemic heart disease).

### 4.4. Follow-Up

The first day of radiotherapy was defined as day 1. Bimanual pelvic examination was performed every 3 months up to 2 years. Pelvic MRI was performed at 3 months. CT from the neck to the pelvis was performed at 6 and 12 months, and annually thereafter.

### 4.5. Statistics

Survival was estimated using the Kaplan–Meier method. The differences in survival between two groups were examined by the log-rank test. The trends in survival across three or more groups were examined by the log-rank test for trend. All statistical analyses were performed using GraphPad Prism7 (GraphPad Software, San Diego, CA, USA). A *p* value of <0.05 was considered as statistically significant.

## 5. Conclusions

We report the influence of the 2018 revisions to the FIGO staging system on the stage distribution and outcomes of cervical cancer patients treated with definitive radiotherapy in the 3D-IGBT era. In our cohort, the revised staging criteria, especially the introduction of a stage IIIC, markedly changed the stage distribution of the patients. The revised staging system demonstrated a prognostic performance broadly consistent with the former version, with more effective stratification of the stage III substages. These results provide evidence of the utility of the FIGO 2018 staging system in the practice of definitive radiotherapy for cervical cancer patients in the 3D-IGBT era.

## Figures and Tables

**Figure 1 cancers-12-01770-f001:**
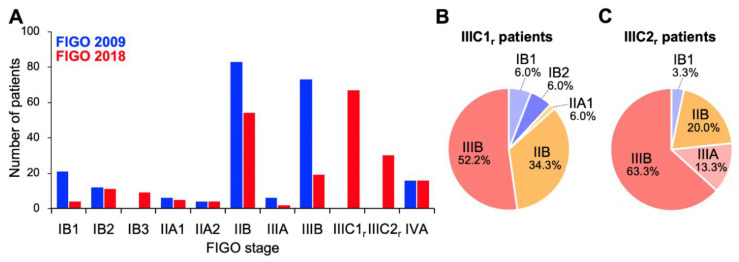
Stage distribution of patients with squamous cell carcinoma of the cervix based on the International Federation of Gynecology and Obstetrics (FIGO) 2009 or 2018 staging criteria. (**A**) Entire cohort (*n* = 221). (**B**) FIGO 2018 stage IIIC1_r_ patients stratified by FIGO 2009 staging criteria (*n* = 67). (**C**) FIGO 2018 stage IIIC2_r_ patients stratified by FIGO 2009 staging criteria (*n* = 30).

**Figure 2 cancers-12-01770-f002:**
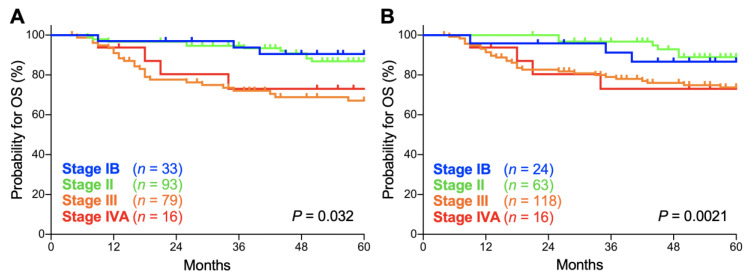
Kaplan–Meier estimates of overall survival (OS) for patients with squamous cell carcinoma of the cervix according to the International Federation of Gynecology and Obstetrics (FIGO) 2009 (**A**) or 2018 (**B**) classification. *p* values calculated from the log-rank test for trend are shown.

**Figure 3 cancers-12-01770-f003:**
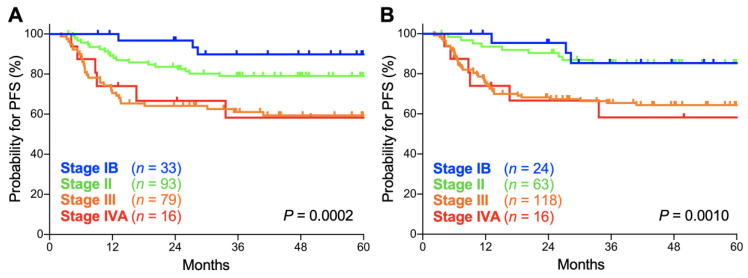
Kaplan–Meier estimates of progression-free survival (PFS) for patients with squamous cell carcinoma of the cervix according to the International Federation of Gynecology and Obstetrics (FIGO) 2009 (**A**) and 2018 (**B**) staging criteria. *p* values calculated from the log-rank test for trend are shown.

**Figure 4 cancers-12-01770-f004:**
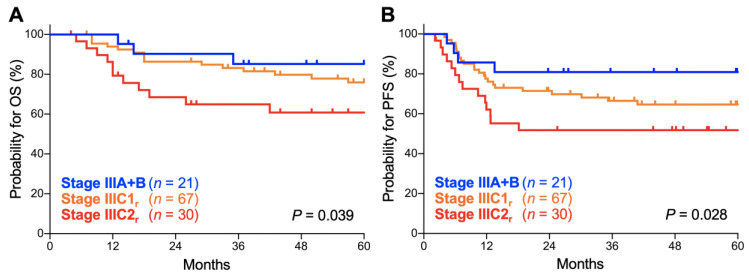
Kaplan–Meier estimates of overall survival (OS) (**A**) and progression-free survival (PFS) (**B**) for stage III patients with squamous cell carcinoma of the cervix according to the International Federation of Gynecology and Obstetrics (FIGO) 2018 staging criteria. *p* values calculated from the log-rank test for trend are shown.

**Figure 5 cancers-12-01770-f005:**
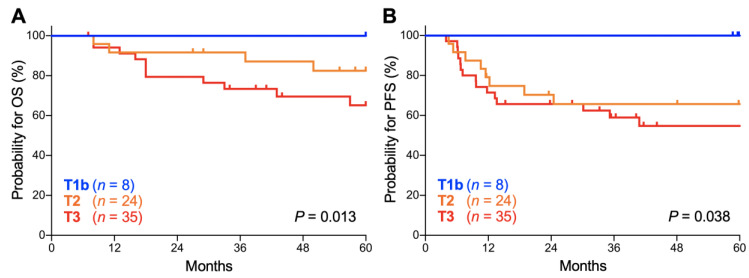
Kaplan–Meier estimates of overall survival (OS) (**A**) and progression-free survival (PFS) (**B**) for stage IIIC1_r_ patients with squamous cell carcinoma of the cervix according to the International Federation of Gynecology and Obstetrics (FIGO) 2018 staging criteria stratified by T stage based on the classification of the Union for International Cancer Control, 7th edition. *p* values calculated from the log-rank test for trend are shown.

**Table 1 cancers-12-01770-t001:** Patient characteristics.

Characteristics	*n* (%)
Age (years)	
<39	14 (6.3)
40–59	91 (41.2)
60–79	103 (46.6)
≥80	13 (5.9)
Treatment	
RT	84 (38.0)
CCRT	137 (62.0)
CT course	
1 or 2	7 (5.3)
3	22 (16.5)
4	42 (31.6)
5	52 (39.1)
6+	10 (7.5)

RT, radiotherapy; CCRT, concurrent chemoradiotherapy; CT, chemotherapy.

**Table 2 cancers-12-01770-t002:** Five-year overall survival for the patients with squamous cell carcinoma of the uterine cervix according to the FIGO 2009 or FIGO 2018 staging schema.

Stage	FIGO 2009	FIGO 2018	*p* Values
IB	90.5 (73.3–96.8)	86.7 (64.1–95.5)	0.65
II	86.8 (77.4–92.5)	88.9 (76.9–94.9)	0.60
III	67.0 (54.8–76.6)	73.7 (64.3–81.0)	0.34
IVA	73.0 (42.9–88.9)	73.0 (42.9–88.9)	1.0

FIGO, International Federation of Gynecology and Obstetrics. Data are % (95% confidence intervals). *p* values calculated from the log-rank test are shown.

**Table 3 cancers-12-01770-t003:** Five-year progression-free survival for the patients with squamous cell carcinoma of the uterine cervix according to the FIGO 2009 or FIGO 2018 schema.

Stage	FIGO 2009	FIGO 2018	*p* Values
IB	89.8 (71.7–96.6)	85.4 (61.1–95.0)	0.63
II	79.0 (69.1–86.1)	85.2 (73.5–92.0)	0.34
III	59.3 (47.3–69.4)	64.3 (54.8–72.3)	0.48
IVA	58.3 (29.1–78.9)	58.3 (29.1–78.9)	1.0

FIGO, International Federation of Gynecology and Obstetrics. Data are % (95% confidence intervals). *p* values calculated from the log-rank test are shown.

**Table 4 cancers-12-01770-t004:** Outcomes of FIGO 2018 Stage III patients with squamous cell carcinoma of the uterine cervix.

Stage	5-Year OS	5-Year PFS
IIIA–B	85.2 (60.8–94.9)	80.9 (56.8–92.3)
IIIC1_r_	75.9 (63.1–84.8)	64.6 (51.6–74.9)
IIIC2_r_	60.8 (40.3–76.1)	51.7 (32.5–67.9)

FIGO, International Federation of Gynecology and Obstetrics. OS, overall survival; PFS, progression-free survival. Data are % (95% confidence intervals).

**Table 5 cancers-12-01770-t005:** Outcomes of FIGO 2018 Stage IIIC1_r_ patients with squamous cell carcinoma of the uterine cervix stratified by tumor T stage.

T Stage	5-Year OS	5-Year PFS
T1b	100.0 (NA–NA)	100.0 (NA–NA)
T2	82.5 (59.7–93.0)	65.6 (42.8–81.1)
T3	65.2 (45.4–79.2)	54.7 (36.0–70.0)

FIGO, International Federation of Gynecology and Obstetrics. OS, overall survival; PFS, progression-free survival; NA, not assessable. Data are % (95% confidence intervals).

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
