# Peer review of "FIGO 2018 Staging for Cervical Cancer: Influence on Stage Distribution and Outcomes in the 3D-Image-Guided Brachytherapy Era"

_cancers, 2020, doi:10.3390/cancers12071770_

Round 1
Reviewer 1 Report
The main research findings of this paper will be important for radiotherapy for SCC of cervical cancer treated by IGBT method. The study provides an important contribution to world-wide gynecologist and radiation oncologist. However, the statistical analyses are poor. And this paper lacks the new findings, for example, whether the number of lymph node metastases affects the prognosis in 2018 classification.
In statistics, the authors used the Kaplan-Meirer method and log-rank test. There are three or more survival curves. Are the results adjusted by Bonferroni correction?
Furthermore, there are some crossing survival curves. It is difficult to discuss the statistical significant differences by log-rank test.
In figures(Fig.2-5), some survival curves resemble and cross. Why all p-values are less than 0.05 ? It is unclear the meaning of these p-values in each figures.
 In Fig.2 and 3,
In stage IB, some cases upgraded higher clinical stage using 2018 classification, especially in IB1 cases. Why the IB cases became worse OS and PFS in 2018 classification?
Author Response
Reviewer 1:
The main research findings of this paper will be important for radiotherapy for SCC of cervical cancer treated by IGBT method. The study provides an important contribution to world-wide gynecologist and radiation oncologist.
Response:
We sincerely thank the reviewer for evaluating our manuscript and for providing insightful comments. According to the comments, we thoroughly revised the manuscript as follows.
However, the statistical analyses are poor.
Response:
We thank the reviewer for the comment. Please see below for the details of the statistical analyses performed in this study.
And this paper lacks the new findings, for example, whether the number of lymph node metastases affects the prognosis in 2018 classification.
Response:
We sincerely thank the reviewer for the insightful comment. We agree with the reviewer that the influence of the number of lymph node metastases on the prognosis in 2018 classification is an important issue that warrants future investigation. Unfortunately, we are not able to provide the relevant data using this cohort due to small number of patients. This was added as the limitation of the study (lines 242–244).
Nevertheless, we believe that our manuscript has the novelty: we for the first time showed the discriminatory ability of the 2018 classification for stage III substages in contrast to the previous studies (references #10, 12, and 13) that contained a considerable amount of old-era data. From this standpoint, our data is the first indication that the 2018 classification is useful for the clinical management of stage III cervical cancers treated with modern-day radiotherapy using 3D-IGBT. This was clarified in the Discussion (lines 208–211 and 217–219).
In statistics, the authors used the Kaplan-Meier method and log-rank test.
Response:
We sincerely apologize. In the method section, we found that we mistakenly noted that "log-rank test" was used to analyze Kaplan-Meier survival curves in Figures 2–5. In reality, we did not use log-rank test; we used "log-rank test for trend". This was corrected (lines 105, 135–136, 159, and 296–297).
There are three or more survival curves. Are the results adjusted by Bonferroni correction?
Response:
We thank the reviewer for the important comment. No, the results were not adjusted by Bonferroni correction. The log-rank test for trend examines the null hypothesis that there is no linear trend between naturally-ordered three or more groups by using chi-square values to calculate a P value. One conduct of log-rank test for trend generates one P value no matter how many groups are being compared. In Figures 2–5, we performed one log-rank test for trend per one figure. Therefore, for these figures, adjustment of alpha errors by Bonferroni correction is not needed.
Furthermore, there are some crossing survival curves. It is difficult to discuss the statistically significant differences by log-rank test.
Response:
We thank the reviewer for the important comment. We agree with the reviewer that some survival curves are crossing in Figures 2–5; this phenomenon is frequently observed for Kaplan-Meier survival estimation for relatively small-sized cohorts as is the case for our study. We also agree with the reviewer that it is difficult to discuss the statistically significant differences for such survival curves by using log-rank test. Therefore, alternatively, we employed the log-rank test for trend; this is a statistical strategy commonly utilized in previous studies (e.g., Cai Z, et al. Cancer Biomarker 2017;19:161–168; Cloney M, et al. World Neurosurgery 2016;89:362–367; Jang RW, et al. Journal of Oncology Practice 2014;10:e335–341; Jenkinson S, et al. Leukemia 2013;27:41–47; McSporran KD. Veterinary Pathology 2009;46:928–933; Nordestgaard BG, et al. JAMA 2007;298:299–308), in which some survival curves are crossing. In addition, the robustness of our statistical strategy was confirmed by Dr. Darwis N.D.M., MD, MPH (MPH degree from Prof. Jill Pell, University of Glasgow, UK), who is an expert in biostatistics.
In Figures 2–5, some survival curves resemble and cross. Why all p-values are less than 0.05 ? It is unclear the meaning of these p-values in each figure.
Response:
We sincerely apologize. In the legends for Figures 2–5, we found that we mistakenly noted that "P values calculated from the log-rank test are shown". In reality, the P values shown in these figures were calculated from the log-rank test for trend. The descriptions were revised thoroughly (lines 105, 135–136, 159, and 296–297). We thank the reviewer for pointing out such critical mistakes.
In Figures 2 and 3, in stage IB, some cases upgraded to higher clinical stage using 2018 classification, especially in IB1 cases. Why the IB cases became worse OS and PFS in 2018 classification?
Response:
We thank the reviewer for the insightful comment. We agree with the reviewer that, in Figures 2 and 3, Kaplan-Meier survival estimates for OS and PFS for stage IB patients are worse in 2018 classification compared with 2009 classification despite upward stage migration in 9 patients. In our dataset, as presented in the Table R1 below, the 9 patients experiencing upward stage migration (highlighted in yellow in Table R1) had favorable prognosis, leading to worse OS and PFS estimates for stage IB patients in 2018 classification. We think that we cannot draw a solid conclusion on the cause of this phenomenon because the number of patients in our dataset was small (n = 33 and 24 in 2009 and 2018 classification, respectively). We added one paragraph to address this as the limitation of this study (lines 233–239).
Please see the attached for Table R1: Raw data of follow-up months and event (i.e., death) for FIGO 2009-stage IB patients with highlights for the patient upgraded to stage III in 2018 classification.

Reviewer 2 Report
It is a very well-researched study and does not require any additional modifications.
Author Response
We sincerely thank the reviewer for evaluating our manuscript and for providing insightful comments.

Reviewer 3 Report
The article is well written, the study design is appropriate and results are clearly explained. I have just a few questions:
1) In table 1 CT course are 4, 5 or others. What do you mean by others?
2) Did you perform all the exams at Gunma University Hospital (CT,MRI,FDG-PET)? Please if yes state it in line 244
3) Lima et al. assesed the prognostic value of 8F-FDG PET/CT in locally advanced cervical cancer (Lima GM, Matti A, Vara G, et al. Prognostic value of posttreatment 18F-FDG PET/CT and predictors of metabolic response to therapy in patients with locally advanced cervical cancer treated with concomitant chemoradiation therapy: an analysis of intensity- and volume-based PET parameters. Eur J Nucl Med Mol Imaging. 2018;45(12):2139-2146. doi:10.1007/s00259-018-4077-1)
Did you perform FDG-PET after treatment? If no why?
Author Response
Reviewer 3:
The article is well written, the study design is appropriate and results are clearly explained. I have just a few questions:
Response:
We sincerely thank the reviewer for evaluating our manuscript and for providing insightful comments. According to the comments, we revised the manuscript as follows.
1) In table 1, CT courses are 4, 5 or others. What do you mean by others?
Response:
We apologize for insufficient description. The majority of the "others" are the patients receiving 3 courses of chemotherapy, and this was mainly due to hematologic toxicities. This was added in the Results (lines 64–65). In addition, Table 1 was revised to clarify the "others" population. We thank the reviewer for the important comment.
2) Did you perform all the exams at Gunma University Hospital (CT, MRI, FDG-PET)? Please if yes state it in line 244.
Response:
We apologize for the lack of such important information. All the pre-treatment examinations were performed at Gunma University Hospital. This was clarified in Diagnostic Modality for Staging of Materials and Methods (lines 269–270). We thank the reviewer for the critical comment.
3) Lima et al. assessed the prognostic value of 8F-FDG PET/CT in locally advanced cervical cancer (Lima GM, Matti A, Vara G, et al. Prognostic value of posttreatment 18F-FDG PET/CT and predictors of metabolic response to therapy in patients with locally advanced cervical cancer treated with concomitant chemoradiation therapy: an analysis of intensity- and volume-based PET parameters. Eur J Nucl Med Mol Imaging. 2018;45(12):2139-2146. doi:10.1007/s00259-018-4077-1). Did you perform FDG-PET after treatment? If no why?
Response:
We thank the reviewer for the valuable comment. Unfortunately, post-treatment FDG-PET was not performed as routine surveillance in our datasets. We agree with the reviewers that FDG-PET/CT has significant roles on post-radiotherapy surveillance. Nevertheless, unfortunately, FDG-PET/CT for routine post-radiotherapy surveillance is not covered by national health insurance systems of Japan. Therefore, we perform FDG-PET/CT for the cases with clinical or radiological suspicion of recurrence by routine bimanual examination, CT, and/or MRI; this strategy is in line with the landmark study of EBRACE II (Potter R et al. Clin Transl Radiat Oncol 2018;9:48–60). This point was described as the limitation of the study (lines 244–247).

Round 2
Reviewer 1 Report
The manuscript has been much improved and is in a nice condition now.
This study is useful for gynecologist and radiation oncologist.